# Modeling Conceptual Framework for Implementing Barriers of AI in Public Healthcare for Improving Operational Excellence: Experiences from Developing Countries

**Sudhanshu Joshi** [1,2], **Manu Sharma** [3,4], **Rashmi Prava Das** [5], **Joanna Rosak-Szyrocka** [6], **Justyna Żywiołek** [6], **Kamalakanta Muduli** [7,*] **and Mukesh Prasad** [2]

1 Operations and Supply Chain Management Research Lab, School of Management, Doon University, Kedarpur 248001, India
2 Australian Artificial Intelligence Institute (AAII), Faculty of Engineering & Information Technology, University of Technology Sydney, Ultimo, NSW 2007, Australia
3 Department of Management Studies, Graphic Era Deemed to be University, Dehradun 248002, India
4 Guildhall School of Business and Law, London Metropolitan University, London N7 8DB, UK
5 Bhubaneswar Engineering College, CV Raman Global University, Bhubaneswar 752054, India
6 Department of Production Engineering and Safety, Faculty of Management, Częstochowa University of Technology, 42-200 Częstochowa, Poland
7 Department of Mechanical Engineering, Papua New Guinea University of Technology, Lae 411, Papua New Guinea
* Correspondence: kamalakanta.muduli@pnguot.ac.pg; Tel.: +675-74272286

**Abstract:** This study work is among the few attempts to understand the significance of AI and its implementation barriers in the healthcare systems in developing countries. Moreover, it examines the breadth of applications of AI in healthcare and medicine. AI is a promising solution for the healthcare industry, but due to a lack of research, the understanding and potential of this technology is unexplored. This study aims to determine the crucial AI implementation barriers in public healthcare from the viewpoint of the society, the economy, and the infrastructure. The study used MCDM techniques to structure the multiple-level analysis of the AI implementation. The research outcomes contribute to the understanding of the various implementation barriers and provide insights for the decision makers for their future actions. The results show that there are a few critical implementation barriers at the tactical, operational, and strategic levels. The findings contribute to the understanding of the various implementation issues related to the governance, scalability, and privacy of AI and provide insights for decision makers for their future actions. These AI implementation barriers are encountered due to the wider range of system-oriented, legal, technical, and operational implementations and the scale of the usage of AI for public healthcare.

**Keywords:** artificial intelligence; healthcare systems; developing countries

## 1. Introduction

Conceptually, the term "Artificial Intelligence" typically refers to a computerized system consisting of hardware, software, and IT infrastructure that aims to perform real-time commercial and non-commercial applications and cognitive functions with structured human inputs [1]. A typical AI-based machine or process runs through mathematical logic and computing programs. A wider variety of methods and claims come under the broader scope of AI, including advanced algorithms, machine learning, deep learning, and pattern recognition [2]. With high productivity and performance, AI has the potential to replace human-oriented work in a wider variety of industrial and social applications. With intensified non-human computational intelligence activities, AI overcomes human limitations [3]. Thus, AI becomes a solution to real-time industrial and managerial problems, ranging from procurement to after-sales services, through various personalized recommendations

to customers through advanced data-driven technologies [3]. Thus, AI tools present a trade-off between potential benefits and risks, as higher risk and greater value through the usage of perceived technologies are preferred over human-centric solutions [3,4]. AI uses supervised and unsupervised machine learning techniques for autonomous decision making for multiple industrial solutions, ranging from BFSI, manufacturing, and retail management to supply chain and logistics management [4,5].

Data-driven services are becoming more AI-driven, and thus, AI has become an important element of business strategies for sustainable competitive advantage [6].

Continuous innovation is bringing new opportunities to various industries, ranging from manufacturing, retail, supply chains, and logistics to transportation and healthcare. A variety of areas in public healthcare leverage the use of artificial intelligence-based technology. Implementing AI in the healthcare system primarily deals with the assessment of the challenges of AI implementation, which aims to manage or alleviate complications and provide ideal treatment for a disease. In public healthcare in particular, AI supports clinical decision support systems for patient-specific diagnosis, treatment decisions, and health analytics [7]. Thus, for the healthcare industry, these AI-driven services are becoming key elements for creating competitive advantages in the ecosystem. On the risk side, public health has several concerns, including potential bias in the data usage for artificial intelligence algorithms, the prevention and protection of patients' privacy, and the healthcare practitioners' distrust of digital tools [7,8].

AI has transformed the delivery of public healthcare in emerging countries for specialty treatments, including radiology and pathology. The digital reform in public healthcare is largely supported by the availability of datasets and the novel methods for assessing these datasets. Despite this, it faces a number of challenges in achieving the SDG public health goals, including the lack of a trained workforce and inadequate public health surveillance systems [9].

Recent advancements in AI have encouraged public enterprises to identify and analyze the risks caused by uncertainty while supporting planning and policy formulation. However, AI interventions requires support from regulators, and practitioners to provide public benefits. AI has been making a lot of progress recently, which is helping public health organizations figure out how bad future outbreaks will be. This is a big improvement over traditional methods, and it will help policymakers and practitioners make better plans and save many lives. However, AI intervention requires that the implementation barriers related to ethics, legality, behavior, and operation are addressed before deployment to developing countries [10].

In this direction, efforts are being made by countries to achieve health-related SDGs among emerging economies. Past research gives evidence to support a variety of health issues addressed by AI, although it also shows the immense requirement to formulate country-specific guidelines and policies for developing an AI implementation roadmap for low-income and emerging economies [11].

The recent developments in the area of AI encourage researchers to investigate and evaluate its adoption and implementation. This study discusses the need for and the significance of AI in public services, especially in technological education, vaccine trials, and data informatics. In particular, in the domain of public health systems, the key usage of AI is in the gathering, diagnosis, and interpretation of medical data. However, in addition to its global adoption and implementation in a variety of industries, there are a few important concerns related to ethical and social issues, including trust and reliability. These issues become prominent as AI-driven healthcare systems carry highly sensitive health information and high-end customer vulnerabilities [11].

In past studies, academic researchers raised key concerns about the implementation of AI-based tools. In developing countries, with the rapid advancements in the area of AI technologies, public organizations are in the process of deploying AI applications to build their productivity and to generate sustainable competitive advantage and value for the beneficiaries.

Leading countries such as India and China are showing a sharp upward trend towards readiness with regard to the adoption of AI applications [12]. The rapid growth of highlighted technologies, such as 3D printing, big data analytics, and ML/DL, brings a collective emergence of industry game changers [13]. AI is envisioned as a critical enabler technology in areas such as public finance, labor markets, marketing, public service advertising, public distribution management, road and transportation, and public information systems [14]. Therefore, AI has become a tool for enhancing the digital capabilities of government in public services and provides a comprehensive socio-economic measure to meet the challenges related to public firms [15]. AI leverages data integration across both inter-organizational and intra-organizational sources and firms to build more customer-oriented, low-cost business solutions [16].

In developing countries, the usage of AI is empowering public services, including enhancing public delivery, precision planning and production, and direct benefit transfer. Technology in its new avatar has the potential to boost economic growth and reduce poverty. The cost of managing public health is very high in developing countries, and it rises during emergency times [17]. In developing economies such as China and India, public health spending has increased by up to 5% of GDP [18]. At the same time, there are opportunities to create value and incentives in health care systems, with the majority of spending on digitalization and automation. Considering the non-availability of a responsive physical health eco-system, in the recent health strategy draft by the WHO a digital health initiative with the effective usage of digital technologies and health informatics was proposed, with the aim of achieving an economic, equitable, and sustainable health system [19]. Thus, the adoption of digital technologies, including smart automation and artificial intelligence, can support the WHO's efforts to make public health more affordable and improve public health. In addition, the constant usage of digital technologies brings a significant impetus to the economic development of the nation [13]. The United Nations (UN) has shown its commitment to aligning multiple stakeholders to evaluate the role and benefits of digital technologies, including AI, to achieve the Sustainable Development Goals (SDGs) [20].

In high-income countries, AI is gradually improving public health services. In the USA, AI applications are saving up to USD 150 billion in healthcare costs [21]. In the context of resource-poor developing countries, the potential of AI in public health needs to be assessed and unleashed [22]. However, developing countries are struggling with two fundamental issues related to the adoption of digital technologies. Firstly, there is the issue of the lack of public health infrastructure and the dearth of trained human resources. It is believed that the new digital wave is creating a psychological fear of unemployment due to the system-level automation of higher cognitive tasks. Such fears provoke mistrust in institutions and rising populist sentiments among the masses [23]. Conversely, due to recent government initiatives and successful public-private partnerships, the Indian healthcare industry is projected to reach to USD 372 billion [24]. On the other hand, the dearth of trained human resources in the healthcare system is facing a global challenge in the imparting of quality health services [25]. The use of digital technologies, including blockchain, BDA, IoT, AR/VR, and artificial intelligence, is helping clinical practitioners to make precision decisions related to the health industry [26]. AI's potential in the healthcare industry is broadly applied in biomedical research, translational research, and medical practice. Thus, AI in health systems amplifies its capabilities [27]. The usage of AI in public healthcare increases operational efficiency and accuracy during diagnosis, in monitoring health conditions, and in reducing surgical complications [28] (Table 1).

**Table 1.** Demonstration of the benefits of using AI in public healthcare.

| [Sr. No] | Benefit of Using AI in Public Healthcare | Description | Developing Countries Perspective | References |
|---|---|---|---|---|
| | | Medical Benefits | | |
| 1 | Data-Driven Decision Making | In medical data processing: acquiring data, analyzing the data, and assessing and evaluating the data for the possible remedies in order to formulate a decision. The staged decision-making process helps medical professionals to understand and make the best use of AI technologies. Thus, AI-based decision systems utilize data in following forms: patient data for clinical decisions; operational data from health centres and hospitals; patients and hospitals to aid in patient decision making | The accuracy and data accessibility determines the quality of decisions in a digital healthcare environment. Particularly in healthcare, the inclusion of smart data helps decision makers to enhance the decision-making quality. Implementation of AI may help the medical statisticans and developers to examine big data. However, managing big data incurs costly processes and impaired clincial outcomes. Thus, continous improvement in data-driven decisions needs to be made. | [13,20] |
| 2 | AI Assistance in surgery | Based on cognitive functions, ML/NPL is being used; it supports touchless arrangements; surgical robots are deployed using speed and voice instruction patterns. AI-surgical robots are computerized equipment; they support surgeons in conducting hands-free surgery. ML enables reinforcement learning that makes the AI-surgical robots access datasets to generate critical data insights and information backups. | In developing countries, the key concern related to surgeon–robot collaborations is, moreover, related to legal and regulatory aspects; the lack of experience of regulatory bodies in dealing with collaborative intelligence is the biggest challenge. LfD schedules are developed to train robots to carry out new surgeries independently through iterative processes that include segmentation of the surgical task, modeling, and subtasking in sequence. Thus, training a machine is again a challenge for developing countries. | [21–23] |
| 3 | AI-assisted tele-surgical operations | Post-surgery requires constant assessment of the patient. Telepresence robots allow surgeons and doctors to interact with their patients for monitoring their vital characteristics without physical presence in the patient's wardroom. | During the COVID-19 pandemic, post-operational tele-surgical operations were used in various developed and developing countries to avoid direct contact between patient and doctors, with intra-operative guidance using remotely accessible videos, pictures, and communication systems. For developing countries, the solution is highly feasible for the places that have poor access to medical health centres and have travel limitations/restrictions due to geography. AI- and AR-enabled surgical mentorship has the potential to become popular among these countries. The key advantages of such arrangements are the systematic minimization of the length of patient stay and the assisting of post-treatment through a remote support system. | [24,25] |
| 4 | Supports mental health | AI-enabled systems for emotional and mental well-being of the patients. | New perspective of AI allows medical practioners to leverage it for understanding mental health of patients | [26] |

**Table 1.** *Cont.*

| [Sr. No] | Benefit of Using AI in Public Healthcare | Description | Developing Countries Perspective | References |
|---|---|---|---|---|
| 5 | Usage of natural language processing | NLP for sentimental analysis. | The healthcare industry conventionally uses natural languague processing for developing computational methods to take human inputs. Sentimental analysis is being used to analyze and interpret vernal expressions of human emotions, including the psychological challenges faced by individual patients. | [27–29] |
| | | | Economic and Social Benefits | |
| 6 | Post-treatment expenditures reduction | Using AI, tailored therapies can be developed for each patient that can bring down the post-treatment expenditures and lower the post-surgery expenditures. | In developing countries such as India, AI facilitates the decisions related to cost optimization, which results in the elimination of expenditure related to post-treatment as the main cost driver in healthcare ecosystem. | [30,31] |
| 7 | Early diagnosis | AI-enabled devices can perform an extensive range of repetitive activities accurately, including the usage of predictive analytics for diagnosis, in order to reduce physician mistakes. | Health precision on electronic health records can bring earlier diagnosis and identification of life-threatening diseases such as breast cancer. The earlier diagnosis can significantly decrease the expenses towards health services. | [32,33] |
| 8 | Empowering patients | AI helps patients to make individual decisions for customized and precision health services. | In recent times, wearable AI devices have become very popular in low-income countries due to their economic range and high acceptability among the masses; machine learning algorithms can help patients to obtain multiple alerts to avoid any serious level of risk. | [34,35] |

The multi-industrial applications and the expanded growth of BDA and AI has prompted the healthcare industry to preview their potential and their risks in public healthcare research [36]. The success rate from other industries has brought growth potential in the healthcare industry as well [33]. Globally, the COVID-19 pandemic has transformed healthcare delivery platforms from conventional face-to-face set-ups to online care using digital tools [37]. While mitigating the risk of exposure to the COVID-19 infection, the healthcare industry has quickly adopted digital collaboration tools for remote clinical aids to patients [38]. According to Accenture (2017), the AI-enabled global healthcare operations have the potential to bring cost optimization to the level of USD 150 billion by 2030 [33]. Various precision health apps are creating clinical–community linkage and establishing dialogue between healthcare providers and patients and catering to the health needs of the masses [39]. In addition, AI also demonstrates the possibility of reducing healthcare costs, providing preventive healthcare to the masses, and increasing the accuracy of diagnoses [40]. Considering this trend, the majority of healthcare solutions companies are adopting scientifically validated AI methods in their R&D projects [41].

*Motivation of the Study*

AI adoption in public sectors and its implementation risks are quite low in some developed countries [42]. In the majority of South Asian emerging economies, the implementation of AI is at an embryonic phase due to a low clarity regarding digital technologies, a lack of AI regulation and laws, and relevant issues pertaining to the data privacy and

trustworthiness among stakeholders [43]. Thus, digitization in public services can be driven by good governance and robust legislation [44,45]. Currently, developing nations are transforming their public systems towards digitization to address the upcoming challenges arising due to the pandemic situation in recent times [46]. In particular, public health systems impose various challenges for emerging economies that are different to those of the developed countries [47]. AI implementation can bring together healthcare solutions for various sections of society, irrespective of the socio-economic status [47]. Thus, there is role for AI in developing countries, where digitally equipped resources and human expertise are very limited and untested. Previous studies have assessed various aspects, including clinical and consumer need identification [48,49] and innovation [50]. AI readiness has challenges [51,52]. A few studies also discussed a framework for the real-time health systems [53], resource optimization [54], and mass usage [55]. There is a need for strong and affective governance and a strategic plan for implementing AI applications in public healthcare, education, and other public sectors [56]. Similarly, recent research has discussed the significance of ethics and policy challenges in the effective governance of AI [57,58]. In the developed countries, AI adoption has witnessed research initiatives, and the efforts increase every day. However, developing countries such as India lack research efforts in the area of AI adoption and its practices. The previous literature suggests that the quality of public health using digital technologies in developing countries has been investigated [55,58]. However, the conceptual frameworks that divulge the inter-relationships among the barriers and enhance AI adoption by effective and appropriate strategies to reduce the implementation barriers are inchoate. In the past few years, digital developments in public domains are highly acknowledge, AI has emerged as the strategic domain for public services [59]. Therefore, to address the societal problems related to public health in emerging economies where the public health systems face a lot of constraints related to capacity planning and operational effectiveness, AI may help local governments to develop ample opportunities [60]. There is an immense need to explore, implement, and expand the usage of AI for public healthcare for the present and future needs of the masses. In a highly populated country such as India, the public healthcare organizations are facing multiple barriers while implementing AI. The past literature shows that the public healthcare industry still expresses few practical concerns about the implementation of artificial intelligence [61]. There is a need for constant research to be carried out with the aim of understanding and evaluating the various implementation challenges of the AI technologies [62,63]. Furthermore, there is a dearth of quality research on the design, development, and implementation of AI-enabled tools to address public health issues [64]. From the AI application perspective, more practical linkages are required to demonstrate the relationship between AI in medicine and consumer care. There is the need to create a strong theoretical foundation for future research on AI implementation for public healthcare. The present study attempts to bridge that gap and aims to address the research inquiries, including the following:

**RQ1**: What are the key implementation barriers to artificial intelligence in public healthcare in the developing countries?

**RQ2**: What is the inter-relationship among the artificial intelligence (AI) implementation barriers in the healthcare industry in the context of developing countries?

**RQ3**: What is the roadmap to reduce the AI implementation barriers in the healthcare industry?

Thus, this study aims to explore the possible solutions to the research questions in the context of AI implementation in developing countries such as India. To enable this purpose, the paper sets the following research objectives:

- To investigate the implementation barriers of AI in public healthcare in developing countries, viz., the Indian context;
- To understand the linkage barriers and the dependent, driving, and autonomous barriers among the selected barriers derived from the systematic literature review (SLR);
- To provide strategic commendations to smoothen the AI implementation in the public health systems.

The paper has the following organization: Section 2 elaborates on the literature on AI technology and public health systems and the theoretical groundwork of the study. The next section explains the methodology and methods used for the present study. Section 4 discusses the detailed research framework on AI implementation in public health. Section 5 explains the key results of the research study. In Section 6, a strategic blueprint is developed to reduce the AI implementation bottleneck in the public healthcare domain. The last section covers the conclusion, limitations, and open research challenges for the future.

## 2. Literature

This section discusses the literature review with respect to artificial intelligence and public health in developing countries. That is presented in this section. The goals of this section are threefold. Firstly, it will explore the AI implementation framework for public healthcare in the context of developing countries. Secondly, past studies are discussed to show the usage of AI implementation through various applications, including facial expression, speech identification using machine learning, and NLP. Thirdly, past conceptual studies are discussed to understand how various industries are using the potential of AI implementation and the integration of AI into the public health systems. Section 2.1, with the supporting literature, deals with the barriers that are affecting AI adoption in public health. Section 2.2 discusses the literature on AI applications in the healthcare industry, and it further discusses the AI disruptive potential; the topics are summarized, and the requirement to proceed with this study is also explained.

Based on "Scopus" and the Web of Science database, a systematic analysis of the literature was carried out on the relevant publications on public health and the usage of AI in this domain. As depicted in Tables 2 and 3, a search protocol was conducted using multiple terms: "Public Health" AND "Artificial Intelligence"; "Public Health" AND "Machine Learning"; "Public Health Systems" AND "Industry 4.0" AND "Digital Technologies". A systematic literature review process was followed to evaluate the prominent publications on AI adoption and its implementation challenges and the digital transformation in public healthcare. For the SLR, the selected timeline was 2017–2022; the first search resulted in 669 articles. The omission of duplicates was then made, leaving 518 articles related to the research questions. Conference proceedings, conference papers, and working papers were also excluded. Seventy-five articles were found to be relevant in the context of the research questions. To conclude the selection of papers, a cross referencing approach was employed, and finally, 35 papers were selected.

**Table 2.** Search Protocol.

| Key Dimensions | Description |
|---|---|
| Keyword | "Public Health" AND "Artificial Intelligence" AND "developing countries" |
| Timespan | 2017–2021 |
| Fields | Article title, detailed abstract, and keywords |
| Inclusion Criteria | Publications in Scopus database |
| Exclusion Criteria | Non-English articles |

**Table 3.** Systematic Literature Review (2017–2021).

| Search Terms | Initial Search | First Screening | Second Screening | Third Screening |
|---|---|---|---|---|
| "Public Health" AND "Artificial Intelligence" | 286 | 148 | 33 | 16 |
| "Public Health" AND "Machine Learning" | 217 | 228 | 21 | 12 |
| "Public Health Systems" AND "Industry 4.0" AND "Digital Technologies" | 166 | 142 | 21 | 7 |
| | | | **Total articles** | **35** |

Due to the novel topic, allied technologies, including IE 4.0 and blockchain technology, were also considered to identify the constructs.

### 2.1. Public Healthcare and Digitalization

Public healthcare is evolving as the fastest growing industry for the existing players. The firms involved in the public healthcare supply chains are adopting new business models [65]. The public healthcare sector does not have advanced technology and is lacking in the adoption of analytics tools/methods as compared to other sectors, such as retail and banking [66]. The public healthcare ecosystem is slowly transforming and the major focus is on the improvisation of the value chains [67]. However, due to the operational issues (such as data privacy and security in a cost-effective manner, etc.), the public health industry has been limited with the improvisation [68]. The aging demography in the developing countries is also an emerging challenge; the median-age population needs more customized health services [69].

Therefore, over the last few years, developing a customer-focused and effective public healthcare system has become an important challenge. AI has the capacity to transform the public health systems. By 2024, AI-based applications in various consumer-centric industries will grow by USD 20 billion, in comparison to USD 315.9 million in 2015 [70]. AI-enabled digital health or electronic health is an evolving area that includes EHRs that aim to provide public services through web-enabled database management [71]. AI-based health systems ensure improvement in public health at the local, regional, national, and global level [72].

However, in developing countries such as India and China public health is in the stage of constant evolution [73]. The inclusion of AI can strategically eliminate the medical errors, which otherwise can be threatening to a patient's health and overall well-being in the healthcare systems [66]. As part of the ICT priorities, implementing the AI-enabled health data is becoming significant [74]. Thus, the adoption of AI in the public healthcare system could be the best practice and learning experience for the global healthcare industry.

### 2.2. Artificial Intelligence and Public Healthcare Systems in Developing Countries

As businesses embrace AI solutions, new challenges have emerged in the corporate adoption, utilization, integration, and implementation of AI in emerging markets. Several conceptual studies have addressed the challenges of AI in services [13], personalization [75,76], advertising [77], sales management [78,79], industrial marketing [80,81], automation in business logistics systems [82,83], market research [84], smart warehousing readiness [85], personal assistance [86], tourism management [87], and ethics [88]. Despite the increasing interest, academic contributions to business-related AI in emerging economies remain scant. The institutional environments in developing countries differ vastly from those in developed countries; this creates obstacles and legitimacy issues for AI-power business applications [89]. Hence, we call for more theoretical and empirical studies to tackle the challenges of AI in emerging markets [90].

Based on the systematic literature review (SLR), a variety of challenges were discussed in the past research including ethical issues [91], the adoption challenges of mobile health wearable devices [92], the scaling of the existing disease surveillance system for public access [93], expandability and public data privacy concerns [94], and the lack of health policies and intelligent systems [95,96]. The healthcare industry is growing each day; thus, the existing players need to adopt new business models [94]. Around sixty percent of the healthcare industry is not prepared to adapt to the integrated and collaborated environment and is therefore unable to adopt and implement the analytics. This sector does not have advance technology and lacks in the adoption of analytical tools/methods as compared to different sectors/fields, such as the retail and banking field [92]. The public health ecosystem is slowly transforming, where the major focus is on the improvisation of the value chains through the usage of digital technology [93]. However, due to the operational issues, such as data privacy and its security in a cost-effective manner, etc., the public

health industry has been limited with the improvisation [94]. The aging demography in the developing nations is also an emerging challenge with regard to the median age of the population [95,96]. Therefore, customer-focused and effective public healthcare systems are needed.

For developing countries, the efficiency of a public healthcare system is entirely based on the quality and reliability of the health data of its patients [97]. The vast quantity and accessibility of electronic health information will influence decision making on multiple fronts, including for patients, physicians, healthcare systems, healthcare providers, and regulatory bodies. The standardization of information storage and retrieval will be critical for facilitating the information exchange across these multiple interfaces [98]. In the last few years, AI has been used by the healthcare industry for transforming the public health systems [99]. By 2025, the market of artificial intelligence will grow by USD 191 billion [13]. Digital health or electronic health is an evolving area that includes electronic health records (EHRs), which are aimed at public services through data generated, delivered, or enhanced through the internet and related technologies [100]. E-health ensures improvement in public health at the local, regional, national, and global levels [101]. Due to the multiple healthcare systems, data redundancy and a higher cost of healthcare services arise [102,103]. The concept of electronic health records (EHRs) was conceived in the 1960s at the Mayo Clinic, Minnesota, and later accepted by the healthcare systems of the developed countries [104]. The major properties that define the level of quality of the EHR dataset include: (a) conformance with existing data structural standards; the conformance of the EHRs can be further classified as value conformance, relational conformance, and computational conformance; (b) data completeness; and (c) data plausibility, which determines the data accuracy. It can be further classified as uniqueness plausibility, atemporal plausibility, and temporal plausibility [105]. With the emergence of AI-based technologies, EHRs are becoming standardized as digital medical records for inter- and intra-hospital and inter- and intra-clinic transactions, although the potential of EHRs is still not unleashed by the developing countries due to various adoption and usage challenges and the lack of regional health regulations related to health safety and privacy. However, countries such as India and China are attempting to create convergence for the existing global health regulations with, for example, the Health Insurance Portability and Accountability Act (HIPAA) and the General data Protection Regulation (GDPR) [106]. Based on the quality of the EHR systems and the available stored data, a healthcare organization conducted a variety of analyses for decision making [107]. Functionally, the EHR system is highly formalized information system which allows integration across multiple healthcare providers [108]. EHRs provide multiple advantages, including medical prescriptions, disease management, and a contribution towards lowering medication errors [109]. However, EHRs have several limitations including, a high waiting time, security concerns, and inter-operability [110]. One of the local concerns is the strategic electronic system plan, which is also identified as a key barrier by previous researchers [111]. AI manages EHRs, financial transactions, insurance claims, and underlined transactions. In developing countries such as India, public health is in the stage of constant evolution [112]. The strategic inclusion of AI can eliminate the medical errors, which otherwise can be threatening to the patient's health and the overall well-being of the healthcare systems [113]. As the part of ICT priorities, implementing EMRs alongside AI is gaining importance [114]. Therefore, the adoption of AI in e-health systems could be the best practice and learning experience for the healthcare industry, globally.

The adoption of e-health is slow in developing countries. The major reasons for low adoption are governance, e-healthcare standards and architecture, patient authentication, infrastructure, data privacy issues, legal and ethical issues, and management issues [115]. The unified theory of acceptance [116] and the technology adoption model [117] were applied in the previous research. The AI implementation barriers to the healthcare industry are identified using a systematic literature review (SLA) validated by the group of experts. The artificial intelligence adoption barriers are exhibited in Table 4. Furthermore, Figure 1

depicts the conceptual framework of the implementation barriers of AI in public healthcare. In addition, it illustrates the various tools and applications of AI, through which sustainable operational excellence can be obtained in the public healthcare systems.

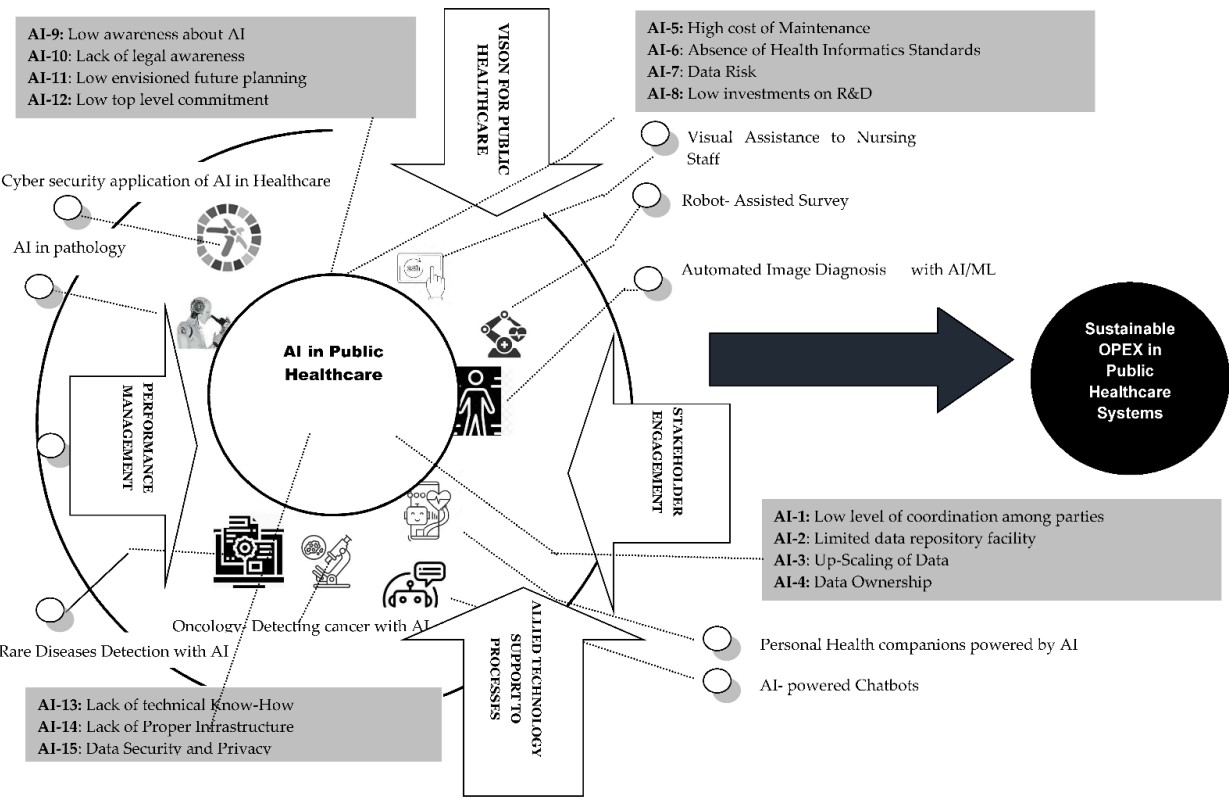

**Figure 1.** Conceptual framework of implementation barriers of AI in public healthcare enhancement.

**Table 4.** AI implementation in public healthcare: key barriers and concerns.

| Code | Implementation Barriers | Description | References |
|------|------------------------|-------------|-----------|
| **AI-1** | Low level of coordination among parties | The synchronization and coordination among various parties including hospital administration, private parties, and suppliers are less and lead to a low level of coordination among parties. | [111,112] |
| **AI-2** | Limited data repository facility | Low scalability to facilitate increasing number of patients beyond a certain capacity. | [113] |
| **AI-3** | Upscaling of data | Due to low level of upscaling of data, real-time data exchange in medical image data storage devices can be disrupted. | [114] |
| **AI-4** | Data ownership | The centralized access of data, which is limited to hospital administration only, reduces the benefits from data reusability. | [116] |
| **AI-5** | High cost of maintenance | Being in the infancy stage, public hospitals are doubtful about the ROCE and ROI of investment in AI implementation. | [116] |
| **AI-6** | Absence of health informatics standards | The global standards for the storage of electronic health records in databases and their retrieval by various AI driven machines are not formed and unified. | [117,118] |
| **AI-7** | Data risk | There are data risk management and security concerns related to AI. | [119] |
| **AI-8** | Low investments on R&D | Low R&D priorities by the public hospitals on health informatics and emerging digital technologies. | [120,121] |
| **AI-9** | Low awareness about AI | The traditional healthcare practitioners are not oriented towards the usage of AI in the healthcare domain. | [121,122] |

**Table 4.** *Cont.*

| Code | Implementation Barriers | Description | References |
|---|---|---|---|
| **AI-10** | Lack of awareness of legal aspects of AI | Low/lack of awareness of legal aspects of implementing AI Creates bottleneck for future upgradation. | [98] |
| **AI-11** | Low envisioned future planning towards technological projects | Due to low vision for future return and non-financial advantages of using AI, the top-level management lacks commitment. | [82] |
| **AI-12** | Low commitment level from top-level management | Low vision and roadmap among top-level management leads to low level of commitment towards implementation of AI. | [123] |
| **AI-13** | Lack of know-how and technical expertise among executives | Due to lack of technical expertise, the implementation stages are adversely impacted. | [124] |
| **AI-14** | Lack of proper infrastructure to support AI implementation | Lack of proper infrastructure leads to low integration between physical and digital ecosystems. | [122] |
| **AI-15** | Data security and privacy | IT infrastructure effectiveness is ensured by high data security and privacy. Thus, weak security may lead to severe privacy issues, including digital theft and fraud. | [122,124] |

## 3. Research Methodology

For the present study, the government hospitals in the state capital have been taken as the case location. The healthcare respondents are key people involved in decision making and policy formulation, top-level managers, and key beneficiaries, including patients from the institutes; this is based on the AI implementation barriers identified from the literature. In recent research, "Multi-Criteria Decision Methods" (MCDM) and "Structuring modeling" has been widely used as an approach aiming to model critical constructs. The methodology uses a limited focus group of experts to obtain the qualitative inputs; the number varies from 12 to 15 experts [125,126]. Interpretive structural modeling (ISM) is employed to identify the inter-relationships amongst the variables and the developing hierarchical structure for the same variables. These methods were used in previous research for identifying adoption barriers in cross-sector collaboration; supply chain flexibility; and developing collaborative intelligent systems [127].

MCDM analysis has been carried out in the recent research to evaluate the AI adoption and implementation in public services, including the public distribution system [128]; the supplier selection for public healthcare [129,130]; pandemic and disaster management [131]; the public manufacturing sector of an emerging economy [132]; and smart healthcare management systems for the selection of healthcare centres [133]. This study applied ISM and fuzzy MICMAC to identify the existing relationships among the various artificial intelligence adoption barriers. The methods are discussed below.

### 3.1. Interpratative Structural Model

This model is used to connect the attributes in a wide-ranging designed model proposed by Warfield (1974). This method comprises a real-time learning mechanism, where groups of elements are actively organized to shape an overall model. The aim of the method is to use the subject knowledge and expertise of the experts to divide the complex systems into several small sub-systems to form a hierarchical structure. The steps include:

Step 1: Identification of implementation barriers from past literature and their validation through consultation with area experts.
Step 2: Establish companionship amongst the implementation barriers.
Step 3: Generate structural self-interaction matrix (SSIM) built on four aspects (V, A, X, O), which represent the direction of the relationship among the implementation barriers.
Step 4: Generate an initial reachability matrix and transitivity check.
Step 5: Develop final reachability matrix and segment the levels.

Step 6: Develop diagraph, and transitive link elimination.
Step 7: Check sum for inconsistency and review the model.

### 3.2. Fuzzy MICMAC Method

This method is very effective in calculating the driving and dependence implementation barriers. The strength of the implementation barriers may vary and be, for example, weak, equal, strong, or not equal. The following are the steps for the fuzzy (Matrice d'Impacts Croisés Multiplication Appliquée a un Classement) MICMAC application.

Step 1: Establish matrix for binary direct reachability from ISM variables. The diagonal values are replaced with zero and transitivity is ignored.
Step 2: Matrix for the fuzzy binary direct relationship, based on fuzzy set theory; the responses are undertaken by the experts.
Step 3: Matrix for the fuzzy MICMAC is stabilized. The repetition of the multiplication of the matrix is performed until the values of the driving and dependence powers become constant.

### 3.3. Data Collection

The sources of data collection were public healthcare centers. The demographics of the experts engaged in the research are exhibited in Table 5. The experts comprise system engineers, medical practitioners, IT managers, and data scientists.

**Table 5.** Details of experts.

| Variables | Number of Experts |
|---|:---:|
| GENDER | |
| *Female* | 8 |
| *Male* | 7 |
| AGE | |
| *25–30 years* | 8 |
| *31–35 years* | 3 |
| *36–40 years* | 2 |
| *41–45 years* | 1 |
| *46–50 years* | 1 |
| EDUCATION | |
| *Ph.D.* | 3 |
| *MD/MBSS* | 4 |
| *Postgraduates* | 2 |
| *Graduates (Btech, BSc.)* | 6 |
| EXPERIENCE | |
| *0–5 years* | 4 |
| *6–10 years* | 5 |
| *11–15 years* | 3 |
| *More than 15 years* | 3 |
| ROLE | |
| *System engineers and IT managers* | 4 |
| *Medical practioners* | 3 |
| *Patients* | 2 |
| *Surgeons* | 4 |
| *Data scientists* | 2 |

## 4. Model Applications

The ISM and the fuzzy MICMAC are applied in the manner discussed in the earlier section.

### 4.1. ISM Application

The ISM application includes the matrices, on the basis of the inputs by the subject experts. The table SSIM, IRM, level segmentation, and FRM are developed using the iteration process of the ISM procedure, as mentioned in Section 3.1 and described in Tables 6 and 7. The MATLAB software was used for performing the transitivity. Figure 2 depicts the ISM levels.

**Table 6.** SSIM.

|       | AI-15 | AI-14 | AI-13 | AI-12 | AI-11 | AI-10 | AI-9 | AI-8 | AI-7 | AI-6 | AI-5 | AI-4 | AI-3 | AI-2 | AI-1 |
|-------|-------|-------|-------|-------|-------|-------|------|------|------|------|------|------|------|------|------|
| AI-1  | X     | V     | V     | A     | A     | A     | A    | A    | A    | V    | V    | V    | V    | V    | A    |
| AI-2  | A     | A     | X     | A     | A     | A     | O    | A    | V    | A    | A    | V    | V    |      |      |
| AI-3  | A     | A     | A     | A     | A     | O     | A    | A    | V    | V    | V    | V    |      |      |      |
| AI-4  | X     | V     | V     | A     | A     | A     | A    | A    | V    | V    | V    |      |      |      |      |
| AI-5  | A     | A     | A     | A     | A     | A     | A    | A    | V    | V    | V    |      |      |      |      |
| AI-6  | A     | A     | V     | A     | A     | A     | A    | A    | V    |      |      |      |      |      |      |
| AI-7  | A     | A     | A     | X     | A     | A     | A    | A    |      |      |      |      |      |      |      |
| AI-8  | O     | V     | V     | A     | V     | A     | A    |      |      |      |      |      |      |      |      |
| AI-9  | O     | O     | O     | A     | A     | A     |      |      |      |      |      |      |      |      |      |
| AI-10 | O     | O     | O     | A     | A     |       |      |      |      |      |      |      |      |      |      |
| AI-11 | O     | V     | V     | A     |       |       |      |      |      |      |      |      |      |      |      |
| AI-12 | O     | V     | V     |       |       |       |      |      |      |      |      |      |      |      |      |
| AI-13 | V     | V     |       |       |       |       |      |      |      |      |      |      |      |      |      |
| AI-14 | X     |       |       |       |       |       |      |      |      |      |      |      |      |      |      |
| AI-15 |       |       |       |       |       |       |      |      |      |      |      |      |      |      |      |

**Table 7.** Initial reachability matrix.

|       | AI-1 | AI-2 | AI-3 | AI-4 | AI-5 | AI-6 | AI-7 | AI-8 | AI-9 | AI-10 | AI-11 | AI-12 | AI-13 | AI-14 | AI-15 |
|-------|------|------|------|------|------|------|------|------|------|-------|-------|-------|-------|-------|-------|
| AI-1  | 1    | 0    | 1    | 1    | 1    | 1    | 1    | 0    | 0    | 0     | 0     | 0     | 1     | 1     | 1     |
| AI-2  | 1    | 1    | 1    | 1    | 0    | 0    | 1    | 0    | 0    | 0     | 0     | 0     | 1     | 0     | 0     |
| AI-3  | 0    | 0    | 1    | 1    | 1    | 1    | 1    | 0    | 0    | 0     | 0     | 0     | 0     | 0     | 0     |
| AI-4  | 0    | 0    | 0    | 1    | 1    | 1    | 1    | 0    | 0    | 0     | 0     | 0     | 1     | 1     | 1     |
| AI-5  | 0    | 1    | 0    | 0    | 1    | 0    | 1    | 0    | 0    | 0     | 0     | 0     | 0     | 0     | 0     |
| AI-6  | 0    | 1    | 0    | 0    | 1    | 1    | 1    | 0    | 0    | 0     | 0     | 0     | 1     | 0     | 0     |
| AI-7  | 1    | 0    | 0    | 0    | 0    | 0    | 1    | 0    | 0    | 0     | 0     | 1     | 0     | 0     | 0     |
| AI-8  | 1    | 1    | 1    | 1    | 1    | 1    | 1    | 1    | 0    | 0     | 1     | 0     | 1     | 1     | 0     |
| AI-9  | 1    | 0    | 1    | 1    | 1    | 1    | 1    | 1    | 1    | 1     | 0     | 0     | 0     | 0     | 0     |
| AI-10 | 1    | 1    | 0    | 1    | 1    | 1    | 1    | 1    | 1    | 1     | 0     | 0     | 0     | 0     | 0     |
| AI-11 | 1    | 1    | 1    | 1    | 1    | 1    | 1    | 0    | 1    | 1     | 1     | 0     | 1     | 1     | 0     |
| AI-12 | 0    | 1    | 1    | 1    | 1    | 1    | 1    | 1    | 1    | 1     | 1     | 1     | 1     | 1     | 0     |
| AI-13 | 0    | 1    | 1    | 0    | 1    | 0    | 1    | 0    | 0    | 0     | 0     | 0     | 1     | 1     | 1     |
| AI-14 | 0    | 1    | 1    | 0    | 1    | 1    | 1    | 0    | 0    | 0     | 0     | 0     | 1     | 1     | 1     |
| AI-15 | 1    | 1    | 1    | 1    | 1    | 1    | 1    | 0    | 0    | 0     | 0     | 0     | 1     | 1     | 1     |

For every implementation barrier, the reachability element and its predecessor are identified. The implementation barriers which are carrying the same values for the reachability and intersection sets are ranked as top-ordered implementation barriers in the ISM hierarchy. Further iterations are performed for the development of a hierarchical structure (Tables 8 and 9).

**Table 8.** Initial reachability matrix.

| Reachability Set | Antecedent Set | Intersection Set |
|---|---|---|
| 1,2,3,4,5,6,7,13,14,15 | 1,2,3,4,5,6,7,8,9,10,11,12,13,14,15 | 1,2,3,4,5,6,7,13,14,15 |
| 1,2,3,4,5,6,7,13,14,15 | 1,2,3,4,5,6,7,8,9,10,11,12,13,14,15 | 1,2,3,4,5,6,7,13,14,15 |
| 1,2,3,4,5,6,7,13,14,15 | 1,2,3,4,5,6,7,8,9,10,11,12,13,14,15 | 1,2,3,4,5,6,7,13,14,15 |
| 1,2,3,4,5,6,7,13,14,15 | 1,2,3,4,5,6,7,8,9,10,11,12,13,14,15 | 1,2,3,4,5,6,7,13,14,15 |
| 1,2,3,4,5,7 | 1,2,3,4,5,6,7,8,9,10,11,12,13,14,15 | 1,2,3,4,5,7 |
| 1,2,3,4,5,6,7,13,14,15 | 1,2,3,4,6,7,8,9,10,11,12,13,14,15 | 1,2,3,4,6,7,13,14,15 |
| 1,2,3,4,5,6,7,13,14,15 | 1,2,3,4,6,7,8,9,10,11,12,13,14,15 | 1,2,3,4,6,7,13,14,15 |
| 1,2,3,4,5,6,7,8,9,10,11,13,14,15 | 1,2,3,4,5,6,7,8,9,10,11,12,13,14,15 | 1,2,3,4,5,6,7,13,14,15 |
| 1,2,3,4,5,6,7,8,9,10,11,13,14,15 | 8,9,10,11,12 | 8,9,10,11 |
| 1,2,3,4,5,6,7,8,9,11,13,14,15 | 8,9,10,11,12 | 8,9,11 |
| 1,2,4,5,6,7,8,9,10,12 | 8,10,11,12 | 8,10,12 |
| 1,2,3,4,5,6,7,8,9,10,11,13,14,15 | 8,10,11,12 | 8,10,11 |
| 1,2,3,4,5,6,7,8,9,10,11,12,13,14,15 | 10,12 | 10,12 |
| 1,2,3,4,5,6,7,13,14,15 | 1,2,3,4,6,7,8,9,11,12,13,15 | 1,2,3,4,6,7,13,15 |
| 1,2,3,4,5,6,7,14,15 | 1,2,3,4,6,7,8,9,11,12,13,14,15 | 1,2,3,4,6,7,14,15 |
| 1,2,3,4,5,6,7,13,14,15 | 1,2,3,4,6,7,8,9,11,12,13,14,15 | 1,2,3,4,6,7,14,15 |

**Table 9.** Level segmentation.

| Level Segmentation: Iteration II | | |
|---|---|---|
| Reachability Set | Antecedent Set | Intersection Set |
| 6,13,14,15 | 6,8,9,10,11,12,13,14,15 | 6,13,14,15 |
| 6,8,9,10,11,13,14,15 | 8,9,10,11,12 | 8,9,10,11 |
| 6,8,9,11,13,14,15 | 8,9,10,11,12 | 8,9,11 |
| 6,8,9,10,12 | 8,10,11,12 | 8,10,12 |
| 6,8,9,10,11,13,14,15 | 8,9,11,12 | 8,9,11 |
| 6,8,9,10,11,12,13,14,15 | 10,12 | 10,12 |
| 6,13,14,15 | 6,8,9,11,12,13,15 | 6,13,15 |
| 6,14,15 | 6,8,9,11,12,13,14,15 | 6,14,15 |
| 6,13,14,15 | 6,8,9,11,12,13,14,15 | 6,13,14,15 |
| Reachability Set | Antecendent Set | Intersection Set |
| 8,9,10,11,13 | 8,9,10,11,12 | 8,9,10,11 |
| 8,9,11,13 | 8,9,10,11,12 | 8,9,11 |
| 8,9,10,12 | 8,10,11,12 | 8,10,12 |
| 8,9,10,11,13 | 8,9,11,12 | 8,9,11 |
| 8,9,10,11,12,13 | 10,12 | 10,12 |
| 13 | 8,9,11,12,13 | 13 |
| Level Segmentation: Iteration IV | | |
| Reachability Set | Reachability Set | Reachability Set |
| 8,9,10,11 | 8,9,10,11 | 8,9,10,11 |
| 8,9,11 | 8,9,11 | 8,9,11 |
| 8,9,10,12 | 8,9,10,12 | 8,9,10,12 |

**Table 9.** *Cont.*

| 8,9,10,11 | 8,9,10,11 | 8,9,10,11 |
| --- | --- | --- |
| **Level Segmentation: Iteration V** | | |
| Reachability Set | Antecendent Set | Intersection Set |
| 10,12 | 10,12 | 10,12 |
| 10,12 | 10,12 | 10,12 |

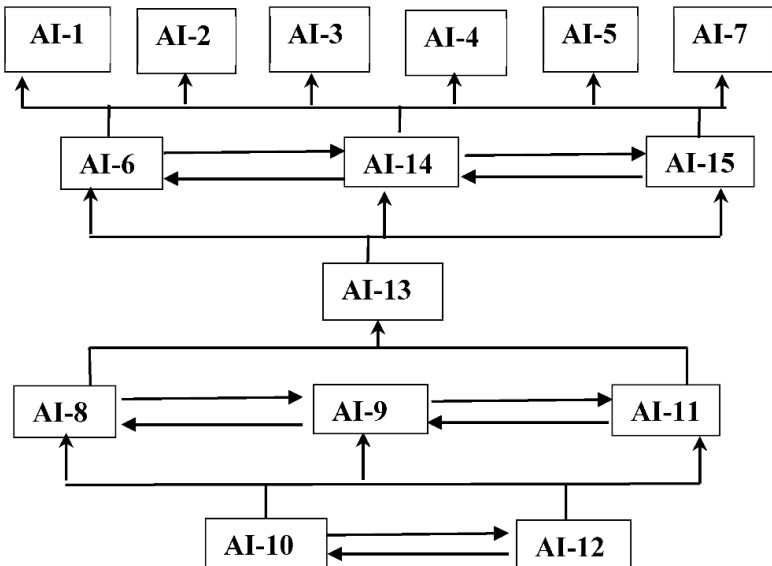

**Figure 2.** ISM levels.

### 4.2. Fuzzy MICMAC Application

The ISM results are undertaken for the application of the fuzzy Matrice d'Impacts Croisés Multiplication Appliquée a un Classement (MICMAC) application. With the step-by-step process discussed, a fuzzy MICMAC stabilized matrix is obtained. Based on the stabilized matrix, four clusters are made: autonomous, dependent, linkage, and driving, and they are presented in Figure 3.

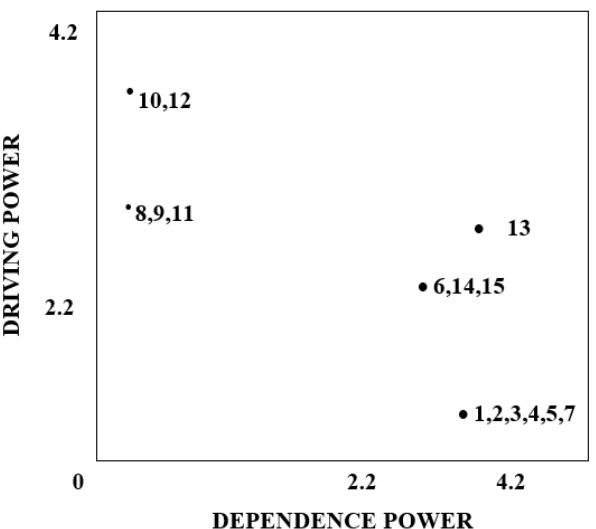

**Figure 3.** Fuzzy MICMAC diagram.

## 5. Findings and Discussion

Based on the SLR, the study initially attempted to address RQ1. Table 4 depicts the key implementation barriers of AI in public healthcare systems, in the context of the developing countries, in order to answer RQ2 and to measure the inter-relationships among the AI implementation barriers in the healthcare industry in developing countries. Based on the expert survey of the subject experts and professionals to assess the hierarchical levels of AI implementation in public healthcare, the integrated ISM Fuzzy MICMAC approach was conducted to formulate the hierarchical level of AI implementation in public healthcare. The ISM results are shown in Figure 2. Low awareness about AI (**AI-9**), lack of awareness of the legal aspects of AI (**AI-8**), low envisioned future planning towards technological projects (**AI-11**), lack of awareness of the legal aspects of AI (**AI-10**), and low commitment from top management (**AI-12**) are the key drivers for all the other AI implementation barriers. Lack of awareness of legal aspects of AI (**AI-10**) and low commitment from top management (**AI-12**) are declared as the implementation barriers that have the maximum driving power and collectively answer RQ1. The results are validated by the fuzzy MICMAC method, which shows the classification of the AI implementation barriers into three main clusters.

**Cluster I** does not have any implementation barrier. This implies that there are no weak implementation barriers in the study.

**Cluster II** represents the dependent barriers; **Cluster III** demonstrates the linkage barriers; **Cluster IV** demonstrates the driving barriers. On the basis of dependence and driving power, the implementation barriers with high dependence and weak driving power are included in **Cluster II**. This cluster includes implementation barriers, low level of coordination among parties, lack of trust (**AI-1**), limited data repository facilities (**AI-2**), upscaling of data (**AI-3**), data ownership (**AI-4**), high cost of maintenance (**AI-5**), data risk (**AI-7**), absence of health informatics standards (**AI-6**), lack of proper infrastructure to support AI implementation (**AI-14**), and data security and privacy (**AI-15**). These implementation barriers require all the other barriers to minimize the impact of the dependent values on the overall performance. There is a lack of know-how and technical expertise among executives (**AI-3**). The linkage implementation barrier is included in **Cluster III** with high driving and dependence power. The driving implementation barriers are included in **Cluster IV**. These barriers have the highest driving and the weakest dependent power. **AI-8**, low investments in R&D, low awareness about AI (**AI-9**), low envisioned future planning towards technological projects (**AI-11**), lack of awareness of the legal aspects of AI (**AI-10**), and lack of commitment from top-level management (**AI-12**) are the driving barriers. These barriers are obtained on the lowest of the ISM levels.

*Practical Implications*

The study outcomes not only contribute to the existing research literature but also give an in-depth understanding of the AI implementation barriers in public healthcare settings in the context of a developing country. The insights from the study can help future researchers and decision makers to understand the significance of the AI implementation barriers and focus on the most critical driving and dependent implementation barriers. From the study, it is clear that the policymakers need to understand the key benefits of AI implementation in the public healthcare. The summarized implications are:

i.   **Securing medical and clinical data**

AI can provide the integration of a variety of partners, suppliers in the healthcare supply chain with shared information from existing information databases, and infrastructure and relevant digital records related to patients, their medical history, and their feedback. The removal of implementation barriers related to data security will be required for the smooth flow of information.

ii.   **Trusted collaboration**

AI can be helpful in reducing data counterfeit and other threats related to healthcare operations. The tracking of a vaccine supply to healthcare systems using AI can be performed for ensuring quality and timely delivery.

iii.    **Holistic quality management**

AI can ensure the holistic quality in the healthcare system. There are various applications of AI, including medical imaging to ensure quality and timely delivery.

As discussed in Section 6, RQ 3 is responded to by the development of a roadmap to reduce the AI implementation barriers to the healthcare industry.

## 6. Strategic Roadmap

The main research aim of the study is to determine the potential of AI in public healthcare and also to evaluate the barriers towards AI implementation in the industry. The research findings can benefit policymakers to develop a strategic roadmap for the implementation of digital technologies (such as BDA, AR/VR, and blockchain technology) in public healthcare. The key outcome of the research is the knowledge about inter-relationship among the barriers related to AI implementation and also the basis of the causality and prominence. Due to various implementation barriers that have an impact on AI implementation, it remains low in emerging economies. In this context, the implementation of AI, particularly for public health, can be enhanced if health systems and policy makers are aware of the barriers that contribute to its successful deployment and have an understanding of the relationships among the implementation barriers. Thus, the research is significant in evaluating the importance of the variety of AI implementation barriers in public healthcare systems. Furthermore, the strategic roadmap consists of the following steps:

- The development of industrial symbiosis leads to a digital ecosystem for resource sharing among parties;
- The development of a centralized AI-enabled system is for the co-creation of new and open healthcare systems;
- The support from the top-level management of key sustainable practices will enhance the focus of the health organizations to collaborate in AI implementation.

### 6.1. Conclusions, Limitations and Open Research Challenges

AI exhibits a great potential to transform the public healthcare sector. If adopted effectively, various operations issues such as public hospital record maintenance costs, inefficient healthcare practices, and data breaching can be easily handled. The overall ability acquired by AI in public healthcare can help hospital and public healthcare centers to fully secure patient data and trails and manage the outbreak of a harmful situation such as that generated by the COVID-19 pandemic. AI in public health is an important area of research to explore collaboration and inter-dependencies. Electronic resource sharing, and public services with support from the government, private organizations, and NGOs can take more initiatives to resolve the ongoing societal issues and provide electronic health services to the public. Very rarely, studies have confirmed the ISM outcomes through fuzzy MICMAC analysis to determine the inter-relationships among the implementation barriers. Our modeling results show the different implementation barriers, according to their significance on scale of dependence, autonomy, linkage, and independence. It has been observed in the interaction with experts that they are still watching the top players utilize the technology to see its impact on their performance and the implications of their day-to-day operations. As the implementation of AI is a costly affair, therefore they do not want to risk it. There is a dearth of quantitative studies about the potential benefits of AI in public health in developing countries. In addition, there is a larger concern related to infrastructural support, functional skills requirement, and implementation readiness among the top-level management. The scarcity of professionals specifically in the combination

of healthcare systems and AI is another concern in developing countries such as India. Top-level management includes either doctors or those with other qualifications.

This technology experiences difficulty in the Indian healthcare sector due to the existing IT act (IT Act, 2000). According to the IT Act, any kind of breach of personal data should compensate the victim. If the healthcare ecosystem can overcome the driving variables, those that are recognized as the barriers to implementing AI in healthcare, then the formulation of a real-time public healthcare system can be realized. Currently, the public healthcare system is having many implementation challenges from data generation to secure and reduce the cost of operations for the two main stakeholders, namely the patients and the healthcare providers. Moreover, AI implementation can be helpful in the identification of low quality and counterfeit drugs and other medical commodities.

### 6.2. Limitations and Future Research Directions

The authors have deployed a multi-criteria decision approach to determine the implementation barriers towards AI adoption in public healthcare. The research carried out in the near future may record the effect of the barriers on AI implementation in the public healthcare sector in the context of developing countries such as India. Furthermore, empirical shreds of evidence can be collected to validate the results of the study, and selected case studies can be prepared. Based on the study and from the public healthcare system perspective, interested parties can develop a collaborative implementation roadmap while designing, managing, and implementing AI across public hospitals and healthcare systems. In addition to the above, various allied technologies, including public clouds, IIoT, and blockchain, are used for data collection, gathering, storage, and access to ensure a secured and decentralized access for effective public healthcare. Future studies can undergo empirical studies for evaluating the AI implementation barriers and their impact on the healthcare industry. In addition to the above, allied technologies, including public clouds, IIoT, and blockchain technology can be useful in data generation and their gathering and data access to ensure operational excellence in public healthcare systems. Further studies can undergo empirical studies for evaluating the AI implementation barriers and their overall impact on the public healthcare industry.

The research limitations can be further improvised in future research works. The identification of implementation challenges and their identification as common entities for both developing and developed countries is difficult. This study evaluated 15 implementation barriers from a single country; thus, for the purpose of generalization, more cross-country data are required. Moreover, more empirical evaluation of the research problem shall be required. Furthermore, various perspectives on the design and development of the conceptual framework can be further expanded and empirically developed from the viewpoint of sustainable public healthcare systems.

**Author Contributions:** Conceptualization, S.J., K.M.; methodology, M.S.; software, S.J.; formal analysis, R.P.D.; investigation, J.R.-S.; resources, J.Ż.; data curation, M.S.; writing—original draft preparation, S.J.; writing—review and editing, K.M.; supervision, M.P. All authors have read and agreed to the published version of the manuscript.

**Funding:** This research received no external funding.

**Institutional Review Board Statement:** Not applicable.

**Informed Consent Statement:** Not applicable.

**Data Availability Statement:** Not applicable.

**Conflicts of Interest:** The authors declare no conflict of interest.

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
