# Peer review of "Modeling Conceptual Framework for Implementing Barriers of AI in Public Healthcare for Improving Operational Excellence: Experiences from Developing Countries"

_sustainability, doi:10.3390/su141811698_

Round 1

Reviewer 1 Report

The study aims to explore AI and its implementation barriers in the healthcare systems in developing countries. The aim of researching AI in healthcare is a significant topic and worths to be shared with academia and practitioners. About the study, I have several comments which require to be involved and corrected before publication:

- Although the work aims to explore "barriers" why that keyword is not involved in the data base search. How do the authors find out the barriers from those papers then?

- How do the authors come up with Figure 1? Does it belong to their findings? If so, it should be explained well, how that figure is set up.

- The reason of selectin the research methodologies should be explained well, why 3.1 and 3.2 methods are selected? Please also explain those methodologies deeply.

- I suggest authors link the research questions with their findings and methods maybe. How are those questions handled and answered at the end?

- Discuss Tables 4-6 in detail. Not clear.

- Please go through the paper and improve English writing more. There are some typos.

After addressing the above comments correctly, the paper can be accepted to be published in Sustainability journal since it is one of significant areas in healthcare subject.

Author Response

the study aims to explore AI and its implementation barriers in the healthcare systems in developing countries. The aim of researching AI in healthcare is a significant topic and worths to be shared with academia and practitioners. About the study, I have several comments which require to be involved and corrected before publication:

Comment 1 : Although the work aims to explore "barriers" why that keyword is not involved in the data base search. How do the authors find out the barriers from those papers then?

Response: Authors are thankful to the reviewer for the valuable comment. The following research contribution has been added in the manuscript [highlighted in red text]

Line no 242-244: A systematic literature review process is followed to evaluate the prominent publication on AI adoption,its implementation challenges, digital transformation in public healthcare”

Line no. 286-289: Based on the Systematic Literature Review (SLR), variety of challenges were discusseed in the past researches including Ethical Issues [112], adoption challenges of mobile health wearable devices [113], scaling the existing disease survillance system of public access [114], Expandability and public data Privacy concerns [115], Lack of Health policy and Intelligent systems [116][117].”

Comment 2 : How do the authors come up with Figure 1? Does it belong to their findings? If so, it should be explained well, how that figure is set up.

Response: Authors are thankful to the reviewer for the valuable comment. The following research contribution has been added in the manuscript [highlighted in red text]

Line no 394-396:Furthermore, Figure 1 depicts the conceptual framework of implementation barriers of AI in public healthcare. In addition, it illustrate various tools and applications of AI through which sustainable operational excellence can be obtained in public healthcare systems”.

 Comment 3 :  The reason of selecting the research methodologies should be explained well, why 3.1 and 3.2 methods are selected? Please also explain those methodologies deeply.

Response: Authors are thankful to the editor for the valuable comment. The following research contribution has been added in the manuscript [highlighted in red text].

Line no. 402-418: “In recent researches, “Multi-Criteria Decision Methods” (MCDM) and “Structuring modeling” has been widely used approach aiming to model critical constructs. Interpretive Structural Modeling (ISM) is employed to identify the inter-relationships amongst variable and developing hierarchical structure for the same. These methods are used in previous research for identifying adoption barriers in cross-sector colalboration[152]; Supply chain flexibility [153]; developing collaborative intelligent systems [154] .MCDM analysis has been carried out in the recent research to evaluate the AI adoption and implmentation in Public services including public distribution system [155]; Supplier selection for public healthcare [156], [157];  for pandemic and disaster management [158]; public manufacturing sector of an emerging economy [159]; smart healthcare management systems for selection of healthcare centres [160]”

Comment 4 : I suggest authors link the research questions with their findings and methods maybe. How are those questions handled and answered at the end?

Response: Authors are thankful to the reviewer for the valuable comment. The following research contribution has been added in the manuscript [highlighted in red text]

Line no. 533-537: “Based on the SLR, the study initially attempt to address RQ1. Table 2 deplicts the key implementation barriers of AI in public healthcare systems, in context to developing countries.  In order to answer RQ2 and to measure the interrelationship among the AI implementation barriers in healthcare industry in developing countries. Based on the expert survey among the subject experts and professionals to assess the hierachical levels of AI implementation in public healthcare”.

Line no 609-630:As discussed in section 6, RQ 3 is responded by developing a roadmap to reduce the AI implementation barriers to the healthcare industry.  The main research aim of the study is to determine the potential of AI in public healthcare and also to evaluate the barriers towards AI implementation in the industry. The research findings can benefit policymakers to develop a strategic roadmap for the implementation of digital technologies (like BDA, AR/VR, and Blockchain technology) in public healthcare. The key outcome of the research is the knowledge about inter-relationship among barriers relalated to AI implementation and also the basis of casuality and prominence. Due to various implementation barriers that have an impact on AI implementation, remains low in emerging economies. In this context, the implementation of AI, particularly for Public Health, can be enhanced if health systems and policy makers are aware of the barriers that contribute to its success deployement and have an understanding of the relationship among the implementation barriers. Thus, the research is significant in evaluating the importance of the variety of AI implementation barriers in public healthcare systems.Futher, The strategic roadmap consist of following steps:

  • Development of Industrial symbiosis leads to digitsal ecosystem for resource sharing among parties;
  • Development of a centralised AI-enabled system for co-creation of new and open healthcare systems
  • Support from top-level management of key sustainable practices will enhance the focus of the health organisation to collaborate for AI implementation”.

Comment 5 : Discuss Tables 4-6 in detail. Not clear.

Response: Authors are thankful to the reviewer for the valuable comment. The following research contribution has been added in the manuscript [highlighted in red text]

Line no 494-498: The ISM application includes the matrices, on the basis of the inputs by the subject experts. The table SSIM, IRM, level segmentation and FRM are developing using the iteration process of ISM procedure as mentioned in section 3.1 and described into table 4 and table 5.  The MATLAB software was used for performing transitivity. Figure 2 depict ISM Levels”

Comment 6 : Please go through the paper and improve English writing more. There are some typos.

Response: Authors are thankful to the reviewer for the valuable comment. A careful examination of English writing and proof reading of the manuscript is done.

After addressing the above comments correctly, the paper can be accepted to be published in Sustainability journal since it is one of significant areas in healthcare subject.

Response: Authors are thankful to the reviewer for words of encouragement

Reviewer 2 Report

The main research objective of the study is to determine the potential of AI in public health and also to assess the barriers to the implementation of AI in industry. The research findings can be useful for policy makers in developing a strategic roadmap for the implementation of digital technologies (such as BDA, AR/VR and Blockchain technology) in public health. A key outcome of the study is knowledge of the interrelationship between barriers to AI implementation and the basis of randomness and familiarity.

The authors used a multi-criteria decision-making approach to determine implementation barriers to the adoption of AI in public health.

The modeling results show the different barriers to implementation, according to their importance on the scale of dependence, autonomy, relationship and independence. It was observed while interacting with the experts that they are still watching the top players use the technology to see its impact on their performance and the importance of their day-to-day operations.

The main shortcoming of the report is that the support of the study is too small - only 15 experts from public health centers. Experts include systems engineers, medical practitioners, IT managers and data scientists. It would be good to at least explain why such a small group is taken into account.

 Additional spell checking and reformatting of figures and tables is required.

Author Response

The main research objective of the study is to determine the potential of AI in public health and also to assess the barriers to the implementation of AI in industry. The research findings can be useful for policy makers in developing a strategic roadmap for the implementation of digital technologies (such as BDA, AR/VR and Blockchain technology) in public health. A key outcome of the study is knowledge of the interrelationship between barriers to AI implementation and the basis of randomness and familiarity.

The authors used a multi-criteria decision-making approach to determine implementation barriers to the adoption of AI in public health.

The modeling results show the different barriers to implementation, according to their importance on the scale of dependence, autonomy, relationship and independence. It was observed while interacting with the experts that they are still watching the top players use the technology to see its impact on their performance and the importance of their day-to-day operations.

Comment 1: The main shortcoming of the report is that the support of the study is too small - only 15 experts from public health centers. Experts include systems engineers, medical practitioners, IT managers and data scientists. It would be good to at least explain why such a small group is taken into account.

Response: Authors are thankful to the reviewer for the valuable comment. The following research contribution has been added in the manuscript [highlighted in red text]

Line no 402-419: In recent researches, “Multi-Criteria Decision Methods” (MCDM) and “Structuring modeling” has been widely used approach aiming to model critical constructs. The methodology uses limited focus group of experts to get the qualitative inputs, the number varies from 12-15 experts [152],[153]. Interpretive Structural Modeling (ISM) is employed to identify the inter-relationships amongst variable and developing hierarchical structure for the same. These methods are used in previous research for identifying adoption barriers in cross-sector colalboration[152]; Supply chain flexibility [153]; developing collaborative intelligent systems [154].MCDM analysis has been carried out in the recent research to evaluate the AI adoption and implmentation in Public services including public distribution system [155]; Supplier selection for public healthcare [156], [157];  for pandemic and disaster management [158]; public manufacturing sector of an emerging economy [159]; smart healthcare management systems for selection of healthcare centres [160]

Line no 450-455

3.3 Data Collection     

The source of data collection was public healthcare centers. The demography of experts engaged in the research is exhibited in Table 3. The experts comprise system engineers, medical practioners, IT managers and data scientist.

Table 3: Details of Experts

Variables

Number of Experts

GENDER

Female

8

Male

7

AGE

25-30 years

8

31-35 years

3

36-40 years

2

41-45 years

1

46-50 years

1

EDUCATION

Ph.D.

3

MD/MBSS

4

Post- Graduation

2

Graduation (Btech, BSc.)

6

EXPERIENCE

0-5 years

4

6-10 years

5

11-15 years

3

More than 15 years

3

ROLE

System engineers and IT Managers

4

medical practioners

3

Patients

2

Surgeon

4

Data scientist

2

 Comment 2: Additional spell checking and reformatting of figures and tables is required.

Response: Authors are thankful to the editor for the valuable comment. The whole manuscript is carefully examined for any possible grammatical and English error.
